# Attitudes of Community Pharmacy Service Users towards Vaccination Programs in Pharmacy: A Cross-Sectional Survey-Based Study in Croatia

**DOI:** 10.3390/pharmacy10060167

**Published:** 2022-12-01

**Authors:** Doris Rusic, Doris Nanasi, Josko Bozic, Anamarija Jurcev Savicevic, Dario Leskur, Ana Seselja Perisin, Darko Modun, Marino Vilovic, Josipa Bukic

**Affiliations:** 1Department of Pharmacy, University of Split School of Medicine, 21 000 Split, Croatia; 2Department of Pathophysiology, University of Split School of Medicine, 21 000 Split, Croatia; 3Teaching Institute of Public Health of Split Dalmatian County, 21 000 Split, Croatia; 4Department of Health Studies, University of Split, 21 000 Split, Croatia

**Keywords:** community pharmacy, vaccination, survey

## Abstract

Background: The aim of this study was to explore community pharmacy service users’ attitudes and opinions towards vaccination programs in pharmacy conducted by a doctor of medicine or a pharmacist. Methods: The questionnaire used in this study comprised 40 items about demographics, sources of information, attitudes about vaccination, attitudes about vaccination in community pharmacies, and willingness to pay for such a service. Results: A total of 385 people participated in this study. Injection was the preferred route of administration of vaccine for more than half of study participants (50.6%). Univariate analysis showed that those who had a healthcare worker as a family member and those familiar with the HPV vaccine had better attitudes; however, those results were no longer significant after factoring in other variables in multivariate analysis. More than half (59.2%) of the study population would consider vaccination service in community pharmacies only if it were free or covered by the national health insurance. Conclusions: More than half of the participants believed that providing vaccination services in community pharmacies would result in greater vaccination rates for seasonal illnesses. However, around half would prefer that it were conducted exclusively by a physician. Less than 10% of the study participants would pay out of their pocket for such a service.

## 1. Introduction

In the recent years, there has been a shift in the focus of community pharmacists. Once focused on medicinal products and drug preparation, today’s pharmacists are, more and more, turning to pharmaceutical care, putting patients at the focus of their work and implementing various pharmacy services [1]. These services may include and are not limited to asthma-related counselling services, monitoring and support on patient adherence to antidepressant medication therapy, anticoagulation and stroke prevention services, assessment of therapy, compliance, lifestyle and social support issues, educational services, pharmaceutical care in the management of diabetes and hypertension in elderly patients, pharmacist-managed repeat dispensing systems, independent prescribing, needle-exchange services, and weight management services [2,3,4].

Pharmacies are the most easily-accessible health care facilities to most patients and are a gateway to other health services to a number of users [5,6]. Furthermore, in many countries, the number of community pharmacies is regulated and, as such, are widely accessible and evenly distributed in the community [7]. This was shown to be a great asset when large scale public health interventions are needed. One example was the need to conduct a widespread vaccination program during the COVID-19 pandemic. Vaccination services were offered in a number of community pharmacies and conducted by physicians on patients with a valid prescription. Previously, a similar service was offered for the seasonal flu-vaccine in selected community pharmacies that had agreed to such service with a physician. Alternatively, other vaccination services are available only through a family physician or the Croatian Institute of Public Health [8].

In the light of a recent announcement on commencing educational programs and trainings for pharmacists to start conducting vaccinations themselves in community pharmacies in Croatia, our aim was to explore community pharmacy service users’ attitudes and opinions on vaccination programs in pharmacies conducted by a doctor of medicine or a pharmacist [9]. Furthermore, we explored users’ opinions on financing such a service or willingness to pay for such service.

## 2. Materials and Methods

### 2.1. Ethical Considerations

This research was approved by the Ethics Committee of University of Split School of Medicine. Notice was given at the beginning of the questionnaire and submitting the questionnaire was considered informed consent for participation in the study. Participation in the research was voluntary and anonymous. Participants received no compensation for participation and could withdraw from the research at any point without penalties.

### 2.2. The Questionnaire

The questionnaire used in this study was designed for the purposes of the study. We conducted an extensive literature search on MEDLINE to identify similar research or research that investigated vaccination practices in pharmacies and users’ attitudes towards vaccination.

The final questionnaire had 40 items. The first part the questionnaire collected demographic data about participants: gender, age, highest education level, average income, and whether the participant or someone from the participant’s family was a health care worker. In the second part, participants were asked if they have a chronic condition, if they had ever received a seasonal flu shot or any other vaccine in adult age, whether they heard about the vaccine against human papilloma virus (HPV), if they knew anyone who has had a severe reaction to a vaccine, and, if they could choose, what their preferred route of administration of a vaccine would be. Also, participants were asked to mark their sources of information about vaccination. In the third part, participants were asked if they have children and if their children received all the obligatory vaccinations from the national vaccination program. Furthermore, they were asked, if they had a child at the time of their answering, would they like their child to receive all the vaccinations from the national vaccination program. Participants were asked if they skipped any of the obligatory vaccines for themselves or their children and what the reasons were for this, if they had. The following part of the questionnaire comprised 19 statements about vaccines that needed to be graded on a 5-point Likert scale, with 1 marking “I fully disagree” and 5 marking “I fully agree with the statement.” In this part, there was a remark that these statements refer to vaccines other than vaccines for the SARS-Cov-2, as we were under the impression that inclusion of these vaccines would yield biased results. In the eyes of the public, vaccines for SARS-Cov-2 were expedited and, as such, do not enjoy great trust from many individuals regarding their safe use [10]. Furthermore, the earliest approved vaccines were developed on a novel platform, further increasing suspicion and raising questions about the safety of the vaccines. In the last part of the questionnaire, participants’ attitudes about vaccination in community pharmacies were evaluated through 5 statements graded on a 5-point Likert scale. These statements considered who would be preforming the service, whether they should be compensated, and how would this service affect the overall vaccination rates in the community. The final question was multidimensional, investigating both coverage and willingness to pay; more precisely, would they pay and how much would our study participants be willing to pay out of their pocket for a vaccination service at a community pharmacy. Possible answers were to enter a certain amount, “I would consider it only if it were free of charge or covered by the national health insurance,” “I get vaccines only at the doctor’s office,” and “I don’t want to get vaccinated.” The final questionnaire was pilot-tested for readability and length among 10 individuals. Following this test, minor language changes were implemented.

### 2.3. Sample Size

In 2021, there were 3,888,529 people in Croatia, all of which were considered possible community pharmacy service users. According to a sample size calculator, with a margin of error set at 5%, the needed sample size for confidence level of 95% was 384 participants. The questionnaire was prepared in Google Forms and sampling was convenient. The link to the questionnaire was distributed through the personal contacts of authors and their contacts via different means of communication, i.e., WhatsApp and similar apps, and public social networks such as Facebook and different groups in those social networks.

### 2.4. Statistical Analysis

Descriptive statistics were used to calculate numbers and proportions. Where appropriate, results are presented as mean ± standard deviation. Answers rated on the Likert-scale were combined to provide number and proportion for “Somewhat disagree,” fully disagree,” “Not sure,” “Somewhat agree,” or “fully agree,” and were presented as such. Furthermore, each participant was assigned an overall attitude score towards vaccination based on their answers on a part of the questionnaire comprised of 19 statements about vaccines that were graded on a 5-point Likert scale. Total score was a sum of answers, with the answer ‘Fully agree’ being assigned five points if the question was formulated as positive towards vaccination and one point if the question was negative towards it. Likewise, the answer on the opposite side of the Likert scale, ‘Fully disagree,’ was assigned one or five points depending how the question was formulated. Answers in-between were assigned four, three, or two points. A higher score indicated more favorable views towards vaccination, and the score ranged from a minimum of 19 to a maximum of 95 points. Linear regression analysis was performed to determine the factors associated with more positive attitudes. Univariate analysis was performed for each variable, with the attitude score serving as a dependent variable. Multivariate regression analysis was further conducted by including factors that were significantly associated with attitude score in univariate analysis. The multiple level (categorical) variables were entered into the regression analysis as multiple dummy-level variables.

## 3. Results

A total of 385 people participated in this study. There were 305 (79.2%) women and 80 (20.8%) men, with a mean age of 30 years (SD 12 years). Most of the participants finished gymnasium high school education or four- and five-year vocational high school education (40.0%), and most had an average monthly income below the minimum wage (51.4%). There were 75 health care workers among our study participants and 104 had a family member who was a health care worker (Table 1). More than half of the study participants reported that they received a seasonal vaccine in adult age (N = 201, 52.2%), and most of them heard about the vaccine against the Human papillomavirus (HPV) (N = 365, 94.8%). As many as 91 persons (23.6%) reported that they know someone who has had a severe reaction to a vaccine. There were 75 (19.5%) healthcare workers and 104 (27.0%) reported having a healthcare worker in their family. Furthermore, 66 (17.1%) participants reported that they have a chronic illness.

Injection was the preferred route of administration of vaccine for more than half of the study participants (N = 195, 50.6%), while 154 (40.0%) would prefer oral liquid and only 36 (9.4%) would prefer nasal spray. As sources of information about vaccines and vaccination programs, study participants most frequently listed health care workers, in 81.0% of cases (N = 312). This was followed by the internet for around half of the participants, while social networks were sources of information for less than a quarter (Table 1).

A total of 88 (22.9%) participants reported that they have children and 82 (93.2%) of them stated that their children have received all the mandatory vaccinations from the national vaccination programme. As many as 354 (91.9%) participants stated that they would like their children to receive all the mandatory vaccinations if they had a child today. Only 26 persons (6.8%) delayed or skipped vaccination for themselves or their child. Reasons included ongoing illness in 9 (34.6%) cases, allergies in 4 (15.4%) cases, and possible adverse reactions and fear in 2 (7.7%) cases each. Other reasons, each in 1 (3.9%) case, were: vaccination for other disease because of a risky needle stick, therefore the regular vaccine was skipped; the COVID-19 pandemic caused delay; war; potential contraindication in family history; previously had experienced adverse reaction; forgot. Three persons (11.5%) left the reason unanswered.

Most study participants agreed that vaccines are safe and that they represent a great step forward in modern medicine. Moreover, not many agreed with statements that vaccines in the national vaccination program are obsolete because the illnesses are harmless or do not exist in Croatia. Participants were more likely to agree with the statement that they have good knowledge on infectious diseases than on how vaccines work. While participants were likely to agree with the statements that risk groups should be vaccinated, that vaccinations against all infectious diseases should be mandatory for all health care workers, and that it is necessary to receive all mandatory vaccines and to vaccinate children for the health of the community, they were divided on the statements that one person’s right to agree to a medical procedure meant every vaccination should be voluntary and that it is legitimate that the government makes vaccination mandatory (Table 2).

According to our results, more than half of the participants believe that providing vaccination services in community pharmacies would result in greater vaccination rates for seasonal illnesses. Furthermore, they were more likely to trust a physician to conduct such a service, but were not likely to agree with the statement that vaccination performed by a qualified pharmacist is less safe than a vaccination performed by a physician or that vaccination in a community pharmacy is less safe than vaccination at a physician’s office. Most study participants agreed that such a service should be paid for out-of-pocket to the pharmacist or physician providing it (Table 3). When we asked the participants how much they would be willing to pay for vaccination service in a community pharmacy, 228 (59.2%) of them stated that they would consider it only if it were free of charge or covered by the national health insurance, 81 (21.0%) stated that they receive vaccines only at a physician’s office, 48 (12.5%) stated that they did not want to get vaccinated, 11 (2.9%) would pay less than 10€, 7 (1.8%) would pay up to 15€, 5 (1.3%) would pay up to 30€, and 5 (1.3%) would pay more than 30€.

Unsurprisingly, people who were vaccinated with seasonal flu or other vaccines and those who would vaccinate their children with all mandatory vaccines had more positive attitudes towards vaccination, while those who knew someone who had a severe reaction had more negative attitudes. Participants who earned less than minimum wage and those who earn above the national average had better attitudes in comparison to those who earn ed from the minimum wage to the national average. All these results were significant in both univariate and multivariate analyses. Univariate analysis showed that those who had a healthcare worker as a family member and those familiar with the HPV vaccine had better attitudes; however, those results were no longer significant after factoring in other variables in multivariate analysis. Univariate linear regression analysis also showed that those who had a bachelor’s degree and three-year high school education had significantly lower attitudes. However, multivariate analysis showed that only those with the highest level of education had more positive attitudes (Table 4).

## 4. Discussion

Ethical questions about vaccinations always seem to cause a divide. Our results indicate that people believe public health is important and would make vaccinations mandatory for health care workers and children, while they do not think that it is legitimate for the government to make vaccinations mandatory and that their right to agree to a medical procedure may be above the interests of the public in terms of importance. Certain vaccines have been mandatory in Croatia for decades [11]; however, lately, there has been an increase in groups that provide resistance to vaccination. In the light of the recent COVID-19 pandemic, this trend has further increased, likely due to fast-tracked development of the vaccines [10]. During the COVID-19 pandemic, there have been efforts to encourage people to receive the vaccine to protect the public health; however, early common national responses included school and public transportation closures, travel and public gathering restrictions and bans, stay-at-home orders, emergency investments in the healthcare system and social welfare, contact tracing, etc. [12]. Refusing to get vaccinated and follow safety measures prolonged the introduced policies and health risks for individuals as well as causing severe economic strains on nations [13]. This, in extreme cases, may be observed as endangering national safety.

Interestingly, most participants would choose to be injected with a vaccine rather than receive the vaccine as an oral liquid or nasal spray. Non-invasive routes of administration are usually perceived to be more acceptable and are believed to result in greater adherence to vaccines [14]. However, from our study, it seems that the public may have reservations towards new formulations. It is possible that they perceive non-invasive routes of administration as less effective. This would be an interesting topic for further research.

Although vaccination services in community pharmacies have so far been offered only for seasonal illnesses [8], we wanted to explore general attitudes and opinions on vaccinations, as we believe that such services may be of great aid to complete mandatory vaccination programs as well. Our results showed that more than half of the studied population agreed with the statement that vaccination services in community pharmacies should be conducted exclusively by physicians. However, more than half disagreed with the statements that vaccination in community pharmacies or by a trained pharmacist is less safe than vaccination at a physician’s office or by a physician. A study conducted among pharmacists in Croatia sowed that 94.5% felt that they need additional training to provide vaccination services and that the license for such a service should be periodical renewed [15].

The majority of the studied population agreed the health care worker providing vaccination services should be adequately compensated; however, most would consider being vaccinated only if it were free of charge or covered by the national health insurance. Less than 10% of the study participants would pay out of their pocket for such service. This may be somewhat explained with the fact that most also reported low income.

This study is not without limitations. Biased results may come from an unbalanced age of our study population. Most of the participants may be considered young. This is likely connected with the method by which the questionnaire was distributed, as the elderly are less likely to use social media or computers. We may have opted to offer the questionnaire in a community pharmacy; however, this would also potentially lead to biased results as it would include mostly frequent community pharmacy service users that may be in favor of additional community pharmacy services. Our study gave an overview of attitudes of young pharmacy service users that may have young children or be about to have children. Children, more precisely, their parents, may be viewed as one of the target populations for vaccination services in community pharmacies, other than the elderly and sensitive groups that require seasonal vaccines. Moreover, this study included mostly women, and they are, indeed, more frequently the primary caretaker that decides on questions such as children’s health. It is encouraging that, although there has been a trend of refusing vaccinations, more than 93% of the parents in this research would agree to all the mandatory vaccines for their children, while almost 92% of the study participants stated that they would like their children to receive all the mandatory vaccinations if they had a child today. Furthermore, healthcare workers were a source of information for more than 80% of subjects.

There were some surprising findings in this study that may be explained by a somewhat flawed instrument. The fact that as many as 23.6% of the participants reported that they knew someone who has had a severe reaction to a vaccine is not in line with other available data. This can be explained with the fact that the question did not precisely define what a severe reaction would be; therefore, this may include mild fever and other conditions that, in fact, are considered to be normal or mild reactions to vaccines. These numbers may have further been aggravated with experiences with the novel vaccines against the SARS CoV-2, as the authorities requested that all observed reactions be recorded. Another unbalance may be observed in a relatively large proportion of health care workers that participated in this study. This, however, seems not to have influenced our results, as it did not significantly impact how favorable a person is towards vaccination in multivariate analysis. Another somewhat odd finding is the fact that most of the participants reported a personal average monthly income to be below the minimum wage. This can be explained with the likelihood that many students were included in this study. Furthermore, due to specific distribution of the questionnaire, we were unable to calculate the exact response rate. Considering the distribution of the questionnaire partly via personal contacts, it is possible that this may have skewed the results, as is visible in the large proportion of healthcare workers that were included. The issue of public health and vaccines has been heightened with the COVID-19 pandemic and this may have influenced responses even though the subjects were instructed to consider other vaccines when completing the survey. This is likely an unavoidable confounding factor of the responses and results.

It would be interesting to conduct this research again after introducing vaccination services by pharmacists and to evaluate the attitudes of both pharmacists providing the service and community pharmacy service users. The results of such research would be beneficial to find ways to adjust the service to the specific needs of the population.

## 5. Conclusions

More than half of the study participants believed that providing vaccination services in community pharmacies would result in greater vaccination rates for seasonal illnesses. However, around half responded that they would prefer if vaccination services were conducted exclusively by a physician. Less than 10% of the study participants would pay out of their pocket for such a service. This research may provide valuable insight into community pharmacy service users’ opinions on vaccination and vaccination services in pharmacies to policy makers.

## Figures and Tables

**Table 1 pharmacy-10-00167-t001:** Study participants’ characteristics and their sources of information.

	All N = 385
Education level	
Three-year vocational education (CQF level 4.1)	5 (1.3%)
Gymnasium high school education; four- and five-year vocational high school education (CQF level 4.2)	154 (40.0%)
University undergraduate studies; professional undergraduate studies (CQF level 6)	98 (25.5%)
University graduate studies; specialist graduate professional studies; postgraduate specialist studies (CQF level 7)	112 (29.1%)
Postgraduate scientific master’s studies or postgraduate university (doctoral) studies (CQF level 8)	16 (4.2%)
Average monthly income	
Below minimum wage (498€)	198 (51.4%)
Minimum wage–average national income (498–941€)	74 (19.2%)
Average national income—1590€	90 (23.4%)
More than 1590€	23 (6.0%)
Health care workers	75 (19.5%)
Family member health care worker	104 (27.0%)
Has a chronic illness	66 (17.1%)
Sources of information about vaccines and vaccination programs	
Health care workers	312 (81.0%)
Internet	203 (52.7%)
Media	188 (48.8%)
Family and friends	179 (46.5%)
Social networks	92 (23.9%)

Results are presented as whole number (proportion); CQF—Croatian Qualifications Framework.

**Table 2 pharmacy-10-00167-t002:** Degree of agreement with the statements about vaccination.

	Somewhat Disagree or Fully Disagree	Not Sure	Somewhat Agree or Fully Agree
Safety			
Vaccines are safe.	38 (9.9%)	57 (14.8%)	290 (75.4%)
I am worried that some vaccines cause autism in children.	223 (57.9%)	86 (22.3%)	76 (19.8%)
I believe that manufacturers make safe and effective vaccines.	51 (13.2%)	96 (24.9%)	238 (61.8%)
Knowledge			
I have good knowledge about infectious diseases.	50 (13.0%)	96 (24.9%)	239 (62.1%)
I have good knowledge on how vaccines work.	48 (12.5%)	95 (24.7%)	242 (62.9%)
Infection offers better protection than vaccination.	126 (32.7%)	139 (36.1%)	120 (31.1%)
It doesn’t matter if the child is vaccinated several times with different vaccines or with a combination of several vaccines.	141 (36.7%)	187 (48.6%)	57 (14.9%)
I remember vaccines I received as child.	127 (33.0%)	75 (19.5%)	183 (47.5%)
Public health			
Vaccinating children is important for the health of the community.	32 (8.3%)	45 (11.7%)	308 (80.0%)
Vaccinations against all infectious diseases should be mandatory for all health care workers.	89 (23.1%)	73 (19.0%)	223 (57.9%)
Risk groups should be vaccinated with seasonal vaccines, i.e., against the flu	42 (11.0%)	77 (20.0%)	266 (69.1%)
Some vaccines from the mandatory programme are not necessary because those illnesses don’t exist in Croatia anymore, i.e., polio.	272 (70.6%)	71 (18.4%)	42 (10.9%)
Some vaccines from the mandatory programme are not necessary because those illnesses are harmless.	276 (71.7%)	69 (17.9%)	40 (10.4%)
Ethical considerations			
Vaccinations represent a great step forward in modern medicine and mankind.	19 (4.9%)	35 (9.1%)	331 (86.0%)
My right to agree to a medical procedure means that every vaccination should be voluntary.	119 (30.9%)	82 (21.3%)	184 (47.8%)
For the purpose of protecting public health it is necessary to receive mandatory vaccines.	47 (12.2%)	43 (11.2%)	295 (76.6%)
It is legitimate that the government makes vaccination mandatory.	128 (33.2%)	84 (21.8%)	173 (45.0%)
Vaccinating childern			
Children receive too many vaccines.	213 (55.4%)	108 (28.1%)	64 (16.7%)
Children get vaccinated while they are still too young.	257 (66.7%)	72 (18.7%)	56 (14.6%)

Results are presented as whole number (proportion).

**Table 3 pharmacy-10-00167-t003:** Study participant’s opinions on vaccination in community pharmacies.

	Somewhat Disagree or Fully Disagree	Not Sure	Somewhat Agree or Fully Agree
More people will get vaccinated against seasonal diseases, i.e., the flu, if vaccination is available in community pharmacies.	56 (14.6%)	103 (26.8%)	226 (58.7%)
Vaccination in the community pharmacy should be conducted only by a physician.	111 (28.8%)	78 (20.3%)	196 (50.9%)
Vaccination by a qualified pharmacist (who underwent additional training) is less safe than vaccination by a physician.	227 (59.0%)	81 (21.0%)	77 (20.0%)
Vaccination in a community pharmacy is less safe than vaccination at a physician’s office.	224 (58.2%)	77 (20.0%)	84 (21.8%)
Pharmacists or physicians who would provide vaccination service in the community pharmacy should be paid for such service.	41 (10.7%)	57 (14.8%)	287 (74.6%)

Results are presented as whole number (proportion).

**Table 4 pharmacy-10-00167-t004:** Linear regression-derived estimates and 95% CI with dependent variable defined as positive attitude towards vaccination score.

Characteristics	Univariate Analysis, Estimate 95% CI	Multivariate Analysis, Estimate 95% CI
**Sex**		
Male	Reference	
Female	−1.7 (−5.1, 1.7)	
**Education**		
Three-year vocational education (CQF level 4.1)	−21.1 (−32.9, −9.3) ***	0.9 (−10.5, 8.7)
Gymnasium high school education; four- and five-year vocational high school education (CQF level 4.2)	Reference	Reference
University undergraduate studies; professional undergraduate studies (CQF level 6)	−6.5 (−9.9, −3.2) **	−2.5 (−5.2, 0.2)
University graduate studies; specialist graduate professional studies; postgraduate specialist studies (CQF level 7)	−3.2 (−6.4, 0.1)	−0.7 (−3.7, 2.4)
Postgraduate scientific master’s studies or postgraduate university (doctoral) studies (CQF level 8)	6.5 (−0.4, 13.3)	6.4 (0.5, 12.3) *
**Average monthly income**		
below 3750 HRK	Reference	Reference
3750–7091 HRK	−9.5 (−13.0, −5.5) ***	−6.3 (−9.2, −3.3) ***
7091–12,000 HRK	−4.6 (−7.9, −1.3) **	−4.3 (−7.4, 1.2) **
above 12,000 HRK	4.0 (−1.7, 9.7)	0.2 (−4.7, 5.2)
**Are you or your family member a healthcare worker?**		
No	Reference	Reference
Yes	3.6 (0.8, 6.4) *	1.6 (−0.6, 3.7)
**Chronic illness**		
No	Reference	
Yes	−0.3 (−4.0, 3.3)	
**Vaccinated as an adult with seasonal flu or other vaccine?**		
No	Reference	Reference
Yes	9.7 (7.1, 12.3) ***	6.1 (4.0, 8.3) ***
**Ever heard of HPV vaccine?**		
No	Reference	Reference
Yes	8.6 (2.5, 14.8) **	4.2 (−0.4, 8.9)
**Do you know anyone who had a severe reaction to the vaccine?**		
No	Reference	Reference
Yes	−11.7 (−14.8, −8.7) ***	−6.1 (−8.7, −3.6) ***
**If you had children, would you vaccinate them with all mandatory vaccines?**		
No	Reference	Reference
Yes	27.0 (22.8, 31.3) ***	20.4 (16.4, 24.5) ***

Data are presented as unstandardized beta (B) coefficient and 95% confidence interval (95% CI). * *p* < 0.05, ** *p* < 0.01, *** *p* < 0.001

## Data Availability

Data is available from the corresponding author upon reasonable request.

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
