# Peer review of "Attitudes of Community Pharmacy Service Users towards Vaccination Programs in Pharmacy: A Cross-Sectional Survey-Based Study in Croatia"

_pharmacy, 2022, doi:10.3390/pharmacy10060167_

Round 1

Reviewer 1 Report

General Comments

This paper is relevant in today’s current public health environment and increased involvement of pharmacists with vaccines.  There does not seem to be a lot of reports about vaccinations and pharmacists specific to this country.  The issue of public health and vaccines has been heightened with COVID and that may have influenced responses even though the subjects were instructed to consider other vaccines when completing the survey. This likely is an unavoidable confounding factor with the responses and results.

In addition, there are some aspects of the methods that need to be reconciled; some with possible methods/analysis adjustments, others with added explanation, and others with additional limitation disclosure. 

 Specific Comments

INTRODUCTION

It may help readers to expand a bit on how vaccines traditionally have been provided and why/how the “pharmacy” aspect of this is relevant.  Those of us unfamiliar with medical practices in Croatia may not understand how a physician would be present in a pharmacy to give vaccines; in many countries medical clinics are not locations where physicians would be found/expected. That kind of clarification will help frame the survey and responses for readers.

METHODS

It is interesting that the researchers chose to capture demographic information at the beginning of the survey. Usually they are held for the end of surveys (e.g., via Dillman survey technique guidance). Key variables pertinent to the topic, such as whether they (and their family) have been vaccinated, where, and preferred routes where would be appropriate to begin framing the attitudinal items and close with demographic variables (“for categorizing” results). Just a thought for future survey efforts, perhaps.

Some of the survey items seem awkward.  The level of knowledge about infectious diseases and how vaccines work may have been quantified with a better scale than the Likert agree/disagree. Similarly, the notion that vaccinating several times vs. with a combination along with infection offering better protection seem more of “knowledge” assessment items.  Whether one remembers getting vaccinated or not seems like a dual choice, yes/no item; what does different level of agreement reflect for this construct?  Perhaps the items and instrument could be describe (and results presented) with this organization?

The scale includes “not sure’ as a response and presumably it was a middle-value rating. With a Likert agree/disagree scale, “neutral” is an option or “neither agree or disagree” but if the items truly are things to capture opinions about, a middle “neutral” response is argued by some researchers to be unnecessary and problematic.  Fortunately, less than 25% of respondents chose the “not sure” response for the attitude items. There were more “not sure” responses with some items, but they were the “knowledge” items noted above.

Five of the last items on the survey were specific to obtaining vaccines in a community pharmacy (lines 91-97). That should be clear and revising to be more complete about what those items covered (e.g., availability, whom should do it, training, etc.) would help readers. Arguably, the last item may be faulty because it does not distinguish between payment to a pharmacist or physician; it is a dual loaded item. (Or, it should have been worded to focus on paying for vaccinations in a pharmacy regardless of whom gives it – pharmacist or phycisian).

In addition, there was a final item about “financial” aspects of getting a vaccine in a pharmacy. Since these “financial” aspects of deciding to get vaccines in a pharmacy (coverage and willingness to pay) were mixed into one broad item, it would be good to describe that question and responses carefully. The item was specific to decisions about getting a vaccine in a pharmacy and how coverage may influence it and how much consumers would spend. This might better have been broken into multiple items about where they would choose to obtain vaccines (MD office or pharmacy), whether they would get if not covered, and then how much out-of-pocket they would pay (with or without coverage, likely branched items).  Since the survey item covered disparate things, it is imperative that it be described well; “willingness to pay” is only one dimension of this multi-dimensional item/response set.

Much more detail is needed about how the survey was administered. (lines 101-105) What was the sample frame? How were surveys delivered? (Presumably an e-mail “blast” to an audience with survey link; where did the distribution list come from, whom was on it, how was it sent, and reminder notices, etc.) This part of the research methods is too cryptic.

As I understand “univariate” regression analysis, it is akin to ANOVA and not necessarily with substantial advantage. The results should be consistent, but possibly easier digested with omnibus ANOVA results and post-hoc means comparisons to show how groups differed. (A suggestion for authors to consider.)  The multivariate analysis omitted “non-significant” variables from the univariate regression.  I wonder if a full model with all variables followed with a more parsimonious model would be another analytic approach. It appears all the multiple-level (categorical) variables were entered into the regression analyses as multiple dummy-level variables and that probably should be noted in the analysis description.

RESULTS

Is there any way to provide insight into the survey response rate? That is a customary result for survey work that helps readers gauge the applicability and representativeness of the results. (It goes along with a better description of the survey administration noted above.)

Data are reported as individual items and proportions of responses for each scale level. Is there any value in including an “average” rating level? This can be a quick and easy way to see where the strongest attitudes lie; the highest and lowest averages reflect strongest agree/disagree perceptions.  It may be possible to combine agree ratings into counts/percents “positive” and “negative” (omitting the “neutrals”) with the average score reflecting the “strength” of the attitudes. This might simplify the Table presentations and possibly save space in the article.

Table 1 could be streamlined to include Education and Income variables only. The other variables either already are redundantly stated in the text or could be (age, gender, health-care worker/chronic condition). It may be possible to combine Table 2 into Table 1 for these key descriptive variables about the respondents.

Tables 3 and 4 may be simplified if data reporting noted above was done. Organizing variables as thematic ‘groups’ may be a better way to display the results (attitudes, knowledge, etc.)

In Table 5, the education levels do not match those in the response description section/table. Shouldn’t they? If an alternate analytic approach is adopted, Table 5 may be split into an ANOVA results table and a new multivariate regression table. Somewhere in one or both of these results tables, Ns for the categories of respondents would be helpful; some of the categories have few respondents.

The elaboration of results from the regressions does not include any interpretation of coefficients. Is that something that would help explain the findings? Or, is there a problem with interpreting a regression coefficient that shows a higher total attitude score for some group relative to the referent group?

Author Response

Please find attached the comments. Thank you for your review

Reviewer 2 Report

This study carried out an online questionnaire survey to explore community pharmacy service users' attitudes and opinions on vaccination programs in pharmacy conducted by a doctor of medicine or a pharmacist. The authors found that more than half the participants would consider vaccination service in community pharmacy only if it were free. This is interesting. The manuscript is well written.

I only have some minor concerns on it.

They could not provide the un-response rate.

And the questionnaire was distributed through personal contacts. The participants might be highly selective.

Table 5, the index was not indicated in the table, is it odds ratios? or beta parameter? In addition, only one decimal places are enough.

Author Response

(The authors gave the same response as above.)
